# Modelling and Terrestrial Laser Scanning Methodology (2009–2018) on Debris Cones in Temperate High Mountains

**José Juan de Sanjosé-Blasco [1],\***, **Mariló López-González [2]**, **Estrella Alonso-Pérez [3]** and **Enrique Serrano [4]**

1 Department of Graphic Expression, INTERRA Research Institute for Sustainable Territorial Development, NEXUS Research Group: Engineering, Territory and Heritage, University of Extremadura, Avenida de la Universidad s/n, 10003 Cáceres, Spain

2 Department of Mathematics and Computer Science Applied to Civil Engineering, Polytechnic University of Madrid, Calle del Profesor Aranguren 3, 28040 Madrid, Spain; marilo.lopez@upm.es

3 Department of Applied Mathematics, Technical School of Engineering, Comillas Pontifical University, Calle de Alberto Aguilera 25, 28015 Madrid, Spain; ealonso@icai.comillas.edu

4 Department of Geography, PANGEA Research Group: Natural Heritage and Applied Geography, University of Valladolid, Plaza del Campus s/n, 47011 Valladolid, Spain; serranoe@fyl.uva.es

\* Correspondence: jjblasco@unex.es

**Abstract:** Debris cones are a very common landform in temperate high mountains. They are the most representative examples of the periglacial and nival processes. This work studies the dynamic behavior of two debris cones (Cone A and Cone B) in the Picos de Europa, in the north of the Iberian Peninsula. Their evolution was measured uninterruptedly throughout each August for 10 years (2009–2018) using the Terrestrial Laser Scanning (TLS) technique. The observations and calculations of the two debris cones were treated independently, but both showed the same behavior. Therefore, if these results are extrapolated to other debris cones in similar environments (temperate high mountain), they should show behavior similar to that of the two debris cones analyzed. Material falls onto the cones from the walls, and transfer of sediments follows linear trajectories according to the maximum slope. In order to understand the linear evolution of the two debris cones, profiles were created along the maximum slope lines of the Digital Elevation Model (DEM) of 2009, and these profile lines were extrapolated to the remaining years of measurement. In order to determine volumetric surface behavior in the DEMs, each year for the period 2009–2018 was compared. In addition, the statistical predictive value for position (Z) in year 2018 was calculated for the same planimetric position (X,Y) throughout the profiles of maximum slopes. To do so, the real field data from 2009–2017 were interpolated and used to form a sample of curves. These curves are interpreted as the realization of a functional random variable that can be predicted using statistical techniques. The predictive curve obtained was compared with the 2018 field data. The results of both coordinates (Z), the real field data, and the statistical data are coherent within the margin of error of the data collection.

**Keywords:** Picos de Europa; debris cones; surface dynamic; mathematical modelling; terrestrial laser scanning

## 1. Introduction

Taluses and debris cones are very common slope forms in temperate mountains and constitute one of the fastest sediment transference systems (Figure 1). The materials come from walls and channels, and the processes involved are highly varied. As a result, the way they function is not well understood.

Studies into debris taluses began in the French Pyrenees and the Arctic [1–3], but concerns about their genesis and evolution have been studied in many temperate mountains [4–10]. The main processes analyzed that are involved in the debris dynamic are rockfall, snow avalanches, and debris flows, but surface processes such as creep, rolling, solifluction, physical and chemical weathering, and surface runoff are also present [5,6,11–16]. The current interpretation of the dynamics of active taluses and debris cones is framed within paraglacial environments that favor mass displacements, rockfalls, and modifications of debris taluses [17].

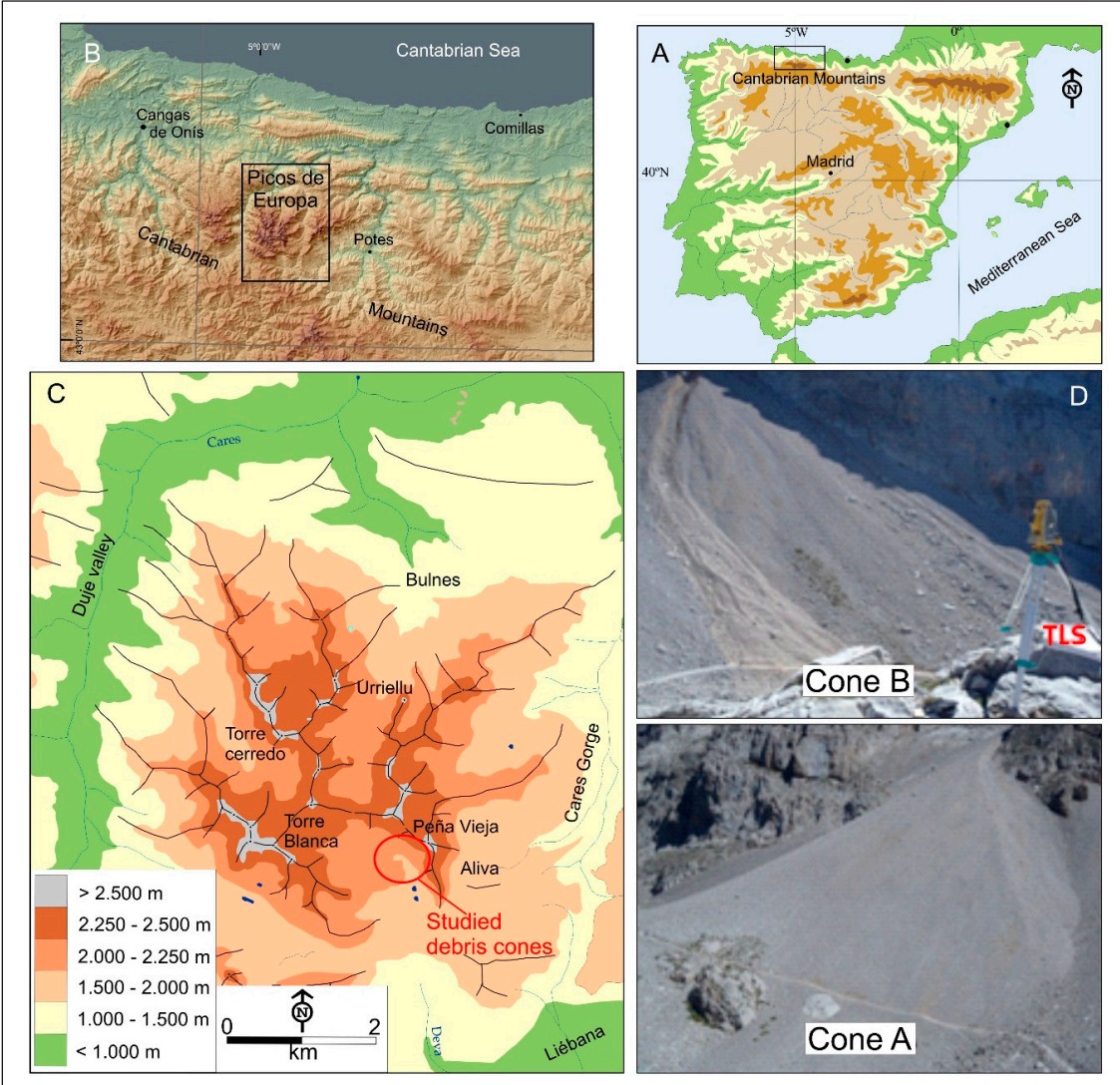

**Figure 1.** Location of the study area and debris cones: (**A**) Cantabrian Mountains in the north of Iberian Peninsula. (**B**) Location of the Picos de Europa massifs. (**C**) The Central Massif of Picos de Europa and location of debris cones in the south face of Peña Vieja. (**D**) Detailed photographs of Cone A and Cone B including terrestrial laser scanning (TLS) station.

The data obtained using different methods for volume and structure measurements of debris cones and taluses may be biased due to the complexity of the system feeding the whole area. Walls and channels, in turn, are a source of materials for transport towards distal areas, with processes that remodel the surface and alter its structure in a cascading sedimentary system [18–22]. These processes are not only geomorphological, such as nivation, debris flows, solifluction, or gelifluction, but also include plant colonization, trampling by animals, and anthropogenic intervention, such as paths and infrastructure, which all contribute alterations with high spatial and temporal variability [23,24].

For the study of debris flow, debris talus, and cone volumes in the temperate mountains of the world, such as the Southern Andes, the Alps, the Pyrenees, the Rocky Mountains, the Carpathian Mountains, the Atlas, the Pindus, the Caucasus, Pamir, or the Zagros, the great potential of remote methods such as Terrestrial Laser Scanning (TLS) and photogrammetry with Unmanned Aerial Vehicles (UAV) is well known [25], and new observation and recording techniques provide more detailed knowledge of the dynamics of taluses and cones [26,27].

Approaches using theoretical models have led to the identification of post-depositional processes in which nivation and debris flows configure the stratified structure [28–30]. The dynamic relationships between walls and taluses and the processes involved have also been analyzed by mathematical formulation [12,16,31,32], and a discrete element model that searches the dynamic of each particle after its fall has been created based on the evolution of the slope of the taluses [32].

The aim of this paper is to further the knowledge of spatial and temporal changes in debris cones as indicators of dominant processes and their dynamic. An attempt is made to discern between the processes of the supply of materials to the upper parts, surface distribution, and distal accumulation through detailed knowledge of the ways in which the surfaces of the debris cones become deformed. It also aims to provide a mathematical formulation of the dynamic of two independent debris cones by modelling to describe the actual situation and to forecast what may happen in the future.

The temporary evolution of the behavior of the debris cones can also be modelled by conducting a stochastic process in continuous time. In this study, functional data analysis techniques were applied for the purposes of prediction, which were framed within statistics with functional instead of discrete data [33–35].

## 2. Study Area

The relief of the Picos de Europa is characterized by its geological structure, the dominance of limestones, karst, and glacial morphogenic systems that, with different rhythms and ages, have shaped a high mountain environment now dominated by nival and periglacial processes. The central massif of the Picos de Europa (Figure 1) reaches 2648 m above sea level (m.a.s.l.), and the proximity of the sea gives it an oceanic climate defined by intense snowfall and precipitation, which surpasses 2500 mm·a−1 at the summits. In the Picos de Europa, the study of periglacial processes has centered on freeze–thaw cycles, the possible existence of mountain permafrost [36–40], and the distribution of active forms and processes such as patterned grounds, ice mounds, solifluction lobes, ploughing blocks or processes associated with ice patches [38,39,41–43], as well as taluses and debris cones [27,37–39,41,44].

The Group of Peña Vieja (2614 m), the third highest mountain group in the Picos de Europa, is located in the central massif and is made up of overlying thrusts to the south, divided by fractures [45] that generate a succession of dorsal ramps with dip of the materials towards the north. The front of the thrust generates abrupt scarps oriented to the south. The local and regional faults and fractures divide this front through fractures in a WNW–ESE direction in blocks raised towards the west. The dominant rocks are limestones from the Namuriense Westfaliense age and Westfaliense Cantabriense age [46,47]. The whole set rises over 400 metres over the surrounding materials generating a continuous scarp 2300 m.a.s.l. from Peña Vieja (2614 m) to the peak Tesorero (2570 m).

Pleistocene glaciations remodeled the Peña Vieja group, configuring hanging glacial cirques, glacial valleys, and moraine complexes belonging to the maximum glacial Pleistocene, such as Aliva and Lloroza, a last phase of advance belonging to the Late Glacial [48,49]. Periglacial processes have remodeled the walls and valley bottoms, conforming broad taluses and debris cones on all its slopes (Figure 2), which are now fully active. However, there are also inactive landforms, such as rock glaciers, solifluction lobes, and stratified debris [37,38], which denote the energetic periglacial activity, possibly in a paraglacial environment, of the last 12,000 years.

The taluses and debris cones of the Picos de Europa are highly representative of the geomorphological dynamic of the massif [37,39,42,50]. There are functional and semi-functional

elements that are distributed between 1200 and 2600 m.a.s.l., with those located over 2000 m.a.s.l. being dominant and fully active [38].

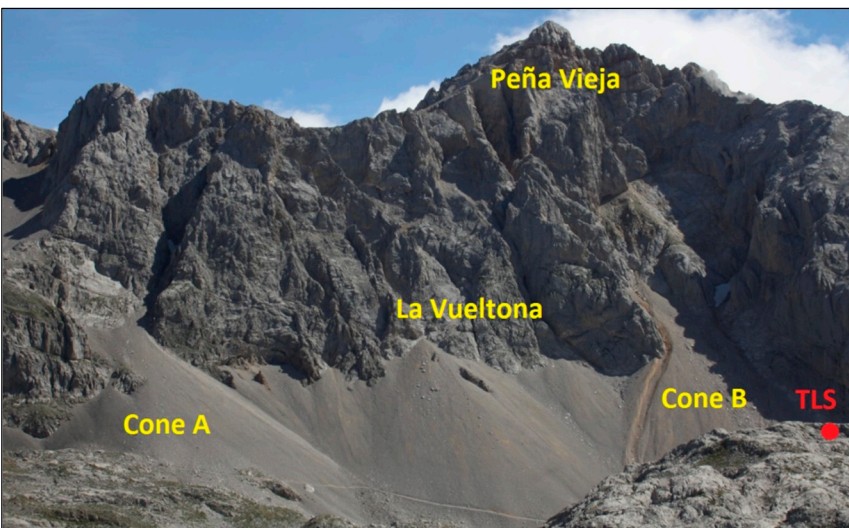

**Figure 2.** General view of the debris cones of La Vueltona in Peña Vieja Group (Picos de Europa Massif).

## 3. Data Collection and Methodology

In the area of La Vueltona, two debris cones were chosen associated with the morphostructural conditions that favored the presence of a wall of over 400 m with unstable materials and with morphoclimatic conditions driving the gravitational, cryogenic, or nival processes to different altitudes. Prior to our studies, geomorphological mapping was available on a scale of 1/25,000 [39], and the detail of the cones and processes of the group of Peña Vieja [27,37], as well as the morphometric study of the set of cones and taluses, were analyzed according to the methodology of Kotarba [7].

### 3.1. Data Collection by TLS

For the dynamic analysis and the surface changes to the debris cones, the Terrestrial Laser Scanning (TLS) geomatic technique was used continuously throughout each August over a period of 10 years (2009–2018). The instrument used was the "Image Station" (Topcon) total station, with the possibility of scanning at distances greater than 1000 m with a positional error of 2 cm from the point measured. The drawback of this equipment is that it measures a different number of points per second depending on the distance. At closer distances (less than 150 m) it measures 20 points per second, but at greater distances it only measures a point every 2–3 seconds. It must be taken into account that conventional TLS, nowadays (2019) at medium range (C10 from Leica or Faro Focus 3D X 330), takes hundreds of thousands of points per second, but their measurement range is limited to 300 m. In our case, the scanning base was on the slope in front of the debris cones 900 m from the furthest point (Figures 1 and 2). From the scanning base, a measurement grid of 3 × 3 m was generated for the two debris cones, and a grid of 40 × 40 cm was generated for the debris flow channel that grooved the left side of cone B. A local system of coordinates was used, but the origin was the same for both cones, though the calculations were processed independently (Figure 3).

From the grid of points, the Digital Elevation Model (DEM) was generated, based on a Triangular Irregular Network (TIN), which allows annual spatial variations in volume to be calculated. Moreover, to know the dynamic behavior of the debris cones, six maximum slope profiles were established (year 2009) in each of the two debris cones (Figure 3). The reason for choosing these profiles was that the rock material that fell from the walls surrounding the debris cones followed the trajectory of a maximum slope line. The dynamic of debris cones was not analyzed superficially since the superficial response was heterogeneous (Figures 13 and 14).

The material in the cones was of heterogeneous size, in general not greater than 30 cm, except in cone B, which had blocks of more than a meter in the distal part (Figure 1). The DEM generated depended on the points measured, and for this purpose the DEM was performed twice in each of the two cones during the same measurement survey (year 2015), and the difference was checked. The mean differences between the two models were 10 cm, and differences did not surpass values of 15 cm in any case except in the distal part of cone B.

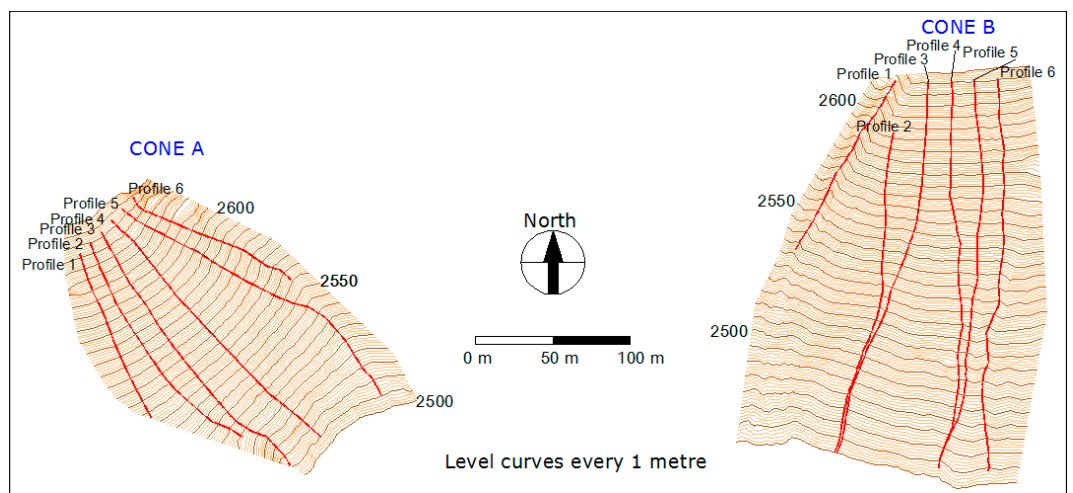

**Figure 3.** Debris cones (A and B) and the representation of the maximum slope profiles over the DEM of 2009.

### 3.2. Dynamic Data Analysis with TLS

The evolution of the cones was calculated by generating tables, with the representation of altitude for distances separated by one meter with respect to a fixed origin. This was performed for each year measured in each of the maximum slope profiles of Figure 3. In general, the total length of each of the profiles was greater than 150 m and the difference in altitude between one annual measurement and the remaining years was not usually more than 30 cm (Figure 4). These were exceptional cases in which the difference was greater than 30 cm (e.g., the distal part of the profiles of cone B), and it was caused by the presence of large blocks of stone (Figure 1). The graphic representation of the altimetric values is expressed in the profiles of maximum slope for each year (Figure 4).

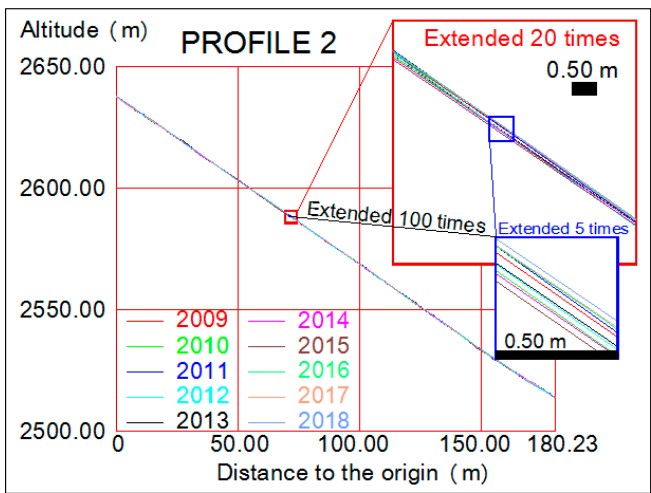

**Figure 4.** Representation of the annual evolution (2009–2018) of profile 2 of cone A (see Figure 3). The blue box is amplified ×100 times with respect to the original profile (red box).

As previously mentioned, for each debris cone 6 maximum slope profiles were determined, and this was done for 10 years of observation; therefore, 120 profiles were available. The mean length of the profiles was 150 m for cone A and 250 m for cone B, so information was available from 24,000 points together with the corresponding altimetric information.

### 3.3. Mathematical Modelling

In order to predict the values of the profiles for the year 2018, data collected annually from the debris cones (2009–2017) were used. For these calculations it was taken into account that, although the data obtained in previous years form a discrete set of values, they are in reality values that belong to the annual curves that represent the evolution of a continuous process. Because of the nature of the phenomenon, the techniques introduced over recent years are considered suitable for the prediction and analysis of functional data [33,35].

These techniques respect the continuous nature of the phenomenon as opposed to the statistical techniques applied to multivariate data. That is to say, the data sample made up of vectors is now formed by curves. For the predictive analysis, an auto-regressive model of the first order, developed by Bosq [33], was applied as provided by the following equation:

$$X_{n+1} = \Psi(X_n) + \varepsilon_{n+1}. \tag{1}$$

where the error $\varepsilon_{n+1}$ and observations $X_n$ are curves, and $\Psi$ is a linear operator that transforms one curve into another. An estimator $\hat{\Psi}$ of operator $\Psi$ will be obtained, fitted thanks to the historical series of the functional sample $\{X_1, X_2, \ldots, X_n\}$, and will provide the prediction of $X_{n+1}$ as $\hat{X}_{n+1} = \hat{\Psi}(X_n)$.

The functional operator $\Psi$ acting on a curve $X$ is considered to be an integral operator:

$$\Psi(X)(t) = \int_0^1 \psi(t,s)X(s)ds \tag{2}$$

in which $\psi(t,s)$ is the kernel of the operator $\Psi$.

The predictive methods in the literature differ among one another in the choice of the operator kernel estimate $\hat{\psi}(t,s)$. After several initial comparisons using different kernel estimators, the EK was chosen, developed in Section 2 of Didericksen [51], which gives very good results.

Some changes must be made to the original sample in order to be able to apply the method mentioned. The initial data sample in our particular case is made up of discrete values obtained annually over the period 2009–2017. The values corresponding to each of these years were distributed uniformly in the interval $[0,1]$ and interpolated by splines to form 9 curves and give the functional sample $\{C_1, C_2, C_3, C_4, C_5, C_6, C_7, C_8, C_9\}$.

As commented in previous sections, work is performed using the differences between consecutive years. Also, the sample must be centered, that is to say, the mean functional sample is subtracted from each curve. In this way, we start out from a functional sample of 8 curves $\{X_1, X_2, X_3, X_4, X_5, X_6, X_7, X_8\}$ with $X_i = (C_{i+1} - C_i) - \frac{1}{8}\sum_{j=1}^{8}\left(C_{j+1} - C_j\right)$, with $i = 1, \ldots, 8$.

From the sample $\{X_1, X_2, X_3, X_4, X_5, X_6, X_7, X_8\}$ we obtain the estimator of the nucleus of the operator $\hat{\psi}(t,s)$, the estimate of the operator $\hat{\Psi}(X)$ and, therefore, the prediction of $X_9$ given by $\hat{X}_9 = \hat{\Psi}(X_8)$.

This methodology may be applied to any of the profiles for which data have been collected. For example, it was applied to profile 3 of cone A (Figures 5 and 6) and profile 3 of cone B, and on these the mathematical modelling was developed.

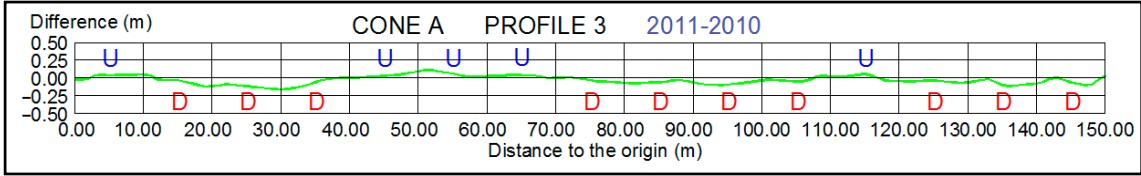

**Figure 5.** Thickening (U) and thinning (D) in the comparison between two consecutive years (2011–2010) of profile 3 of cone A.

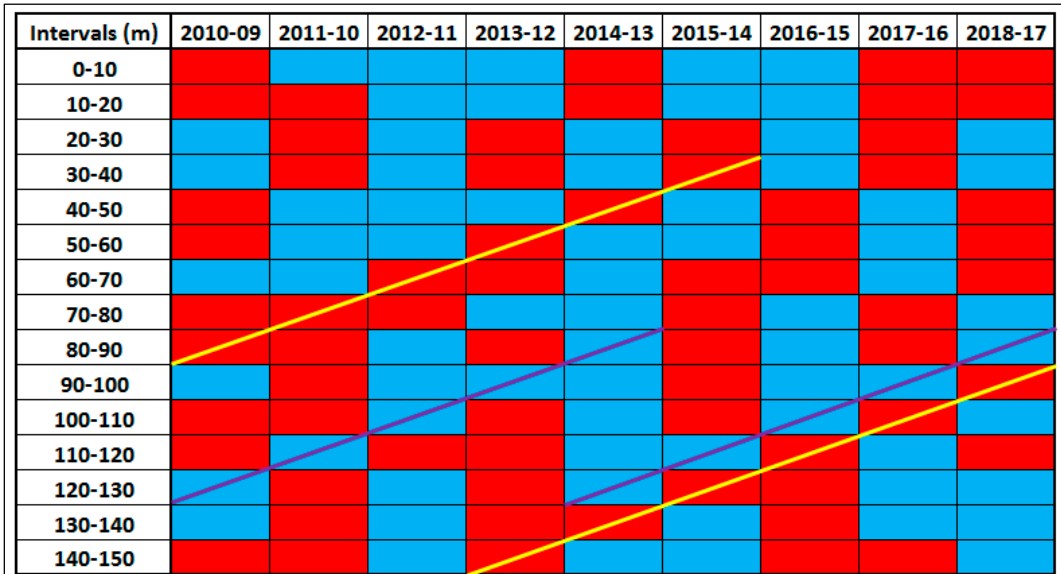

**Figure 6.** Annual evolution of profile 3 of cone A for intervals of distances of 10 m. The values in Figure 5 coincide with the column of the annual period 2011–10. The purple lines correspond to the temporal evolution of thickening of material in the profile, while the yellow lines indicate thinning.

*3.4. Volumetric Analysis*

For the moment, each of the cones has been analyzed using the representation of profiles of maximum slope, but to understand their overall behavior, the set of each debris cones must be analyzed. A DEM has been generated for each observation from the period 2009–2018. These DEMs permit the visualization of the overall changes of each cone and the calculation of the volume (loss and gain of material) in annual periods and in the overall period of 10 years.

In order to be able to compare the changes in volume in each cone, the same outline has been delimited for all years. Therefore, the innermost outline was selected from all the possible annual outlines measured. For the volumetric comparison of the DEMs the following selection of intervals was made (Figures 13 and 14):

- Interval from −0.02 m to 0.02 m; this is the error of measurement generated by the technical characteristics of the instruments;
- Intervals from −0.02 m to −0.15 m and from 0.02 m to 0.15 m; the value of 0.15 m is the maximum difference generated in the DEMs on the same cone and in the same observation survey;
- Intervals from −0.15 m to −0.30 m and from 0.15 m to 0.30 m; up to 0.30 m is the value of the material gain or loss of each of the cones normally in annual periods;
- Intervals from −0.30 m to −1 m and from 0.30 m to 1 m; differences in the measurement on rocks with dimensions close to 1 m;
- Intervals greater than those between −1 m and 1 m; large blocks (greater than a meter). Errors in the edges in generating the DEMs.

## 4. Results

Once the ten DEMs were available (one per year) for the two debris cones for the period 2009–2018, the next steps are the dynamic calculations performed by interpreting the maximum slope profiles using an analogical methodology (spreadsheet representation) (Figure 6) and the analytical technique (mathematical modelling of the dynamic) on one hand, and on the other, studying the general behavior of the debris cones by means of their volumetric analysis.

### 4.1. Calculation of the Slope of the Cones

For each debris cone a section of the central part of each profile (between 50 and 100 m) (Figure 4) was selected. In cone A, the mean slope of the stretch was found to be 66.5% (33.62°), and in cone B it was 66.3% (33.54°). Values close to 33.6° therefore correspond to the equilibrium slope for this kind of rock in cones A and B.

The calculation of the slope at the distal part shows that cone A has a value of 62.1% (31.84°) and cone B 56.8% (29.59°). The difference in the values in each cone in the distal part is due to the different topography of each cone.

### 4.2. Calculation of the Dynamic

To observe the temporal evolution (by annual period) of each of the cones, longitudinal profiles of maximum slope were generated, which were knocked down to a comparison plan for the purposes of analysis. This plan is the origin or the altitude of zero difference (Figure 5).

The partial or annual differences were analyzed, for which purpose two consecutive years were selected, and from these the value of altitude was subtracted from one point of the profile for the year in question, with the value of altitude for the same point of the profile of the year prior to it. Thus, the calculation periods were: 2010–2009, 2011–2010, 2012–2011, 2013–2012, 2014–2013, 2015–2014, 2016–2015, 2017–2016, and 2018–2017.

If the resulting value was positive, it was understood that deposit had been thickened, and if the value was negative it had been thinned. As previously indicated, in general, some values were differentiated by centimeters, and in other exceptional cases the differences were over 30 cm (Figure 5).

Figure 5 shows a profile with distance intervals of 10 m and differential values of altitude with respect to a situation with no differences (0.00 m) compared with the previous year's measurement. If, in the profile, debris material has increased for a 10-m distance interval, it is indicated by the letter U (Up), but if the surface has been thinning, the letter D (Down) is used. This was done for the same profile for annual periods, as in Figure 6.

In Figure 6, lines are drawn of the profile evolution through the years. The color purple shows the evolution of the thickening throughout the profile, and in yellow the thinning. In all cases these were evolutions over five consecutive years.

In both Cone A and Cone B profiles, 2, 3, 4, and 5 were selected (Figure 3). The comparison of 9 annual profiles was made as in Figure 5 for each of the profiles of maximum slope. Therefore, a total of 72 profiles was calculated, each of them as shown in Figure 5.

The behavior of one year with respect to the year immediately before it and the one that followed was analyzed such that years are grouped in threes. For example, the profile of the period 2013–2012 was compared with that of 2014–2013 (Figure 7). Similarly, we proceeded with the rest of the groups of three years, for each profile, as indicated in Figure 7, this is to say, 2014–13 with 2015–14 and 2015–14 with 2016–15.

This study facilitates the analysis of how thinning and thickening are distributed in time and space. In general, the areas of the profile in which a year with thinning is compensated, by thickening, in the following annual period (Figures 6 and 7).

If the altimetric values are subtracted with respect to the value of the first year (2009), the profiles are generated for 2010–2009, 2011–2009, 2012–2009, 2013–2009, 2014–2009, 2015–2009, 2016–2009,

2017–2009, and 2018–2009 (Figure 8). The result of the profile of Figure 8 indicates that there was mainly thickening, and an accumulation of material can be deduced (a greater quantity of blue cells than red ones). This calculation is not significant, given that it is on the profile line, and in order to analyze it as a whole (superficial), a volumetric calculation must be made (Figures 13 and 14).

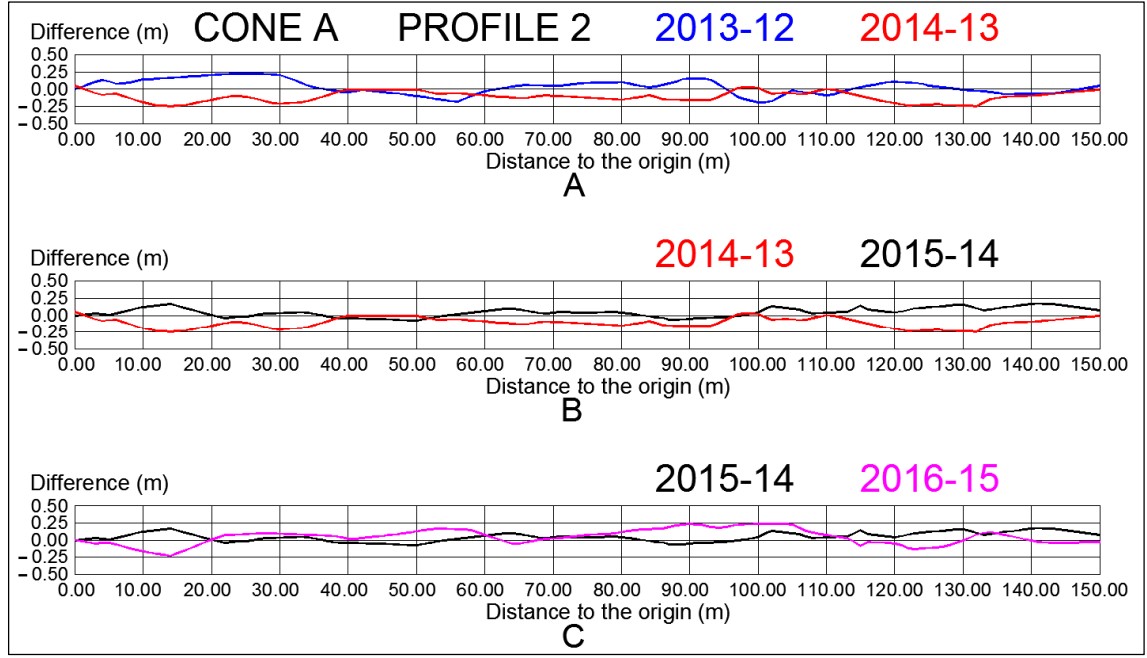

**Figure 7.** Representation profile 2 of Cone A by groups of three years: (**A**) 2013–12 and 2014–13. (**B**) 2014–13 and 2015–14. (**C**) 2015–14 and 2016–15.

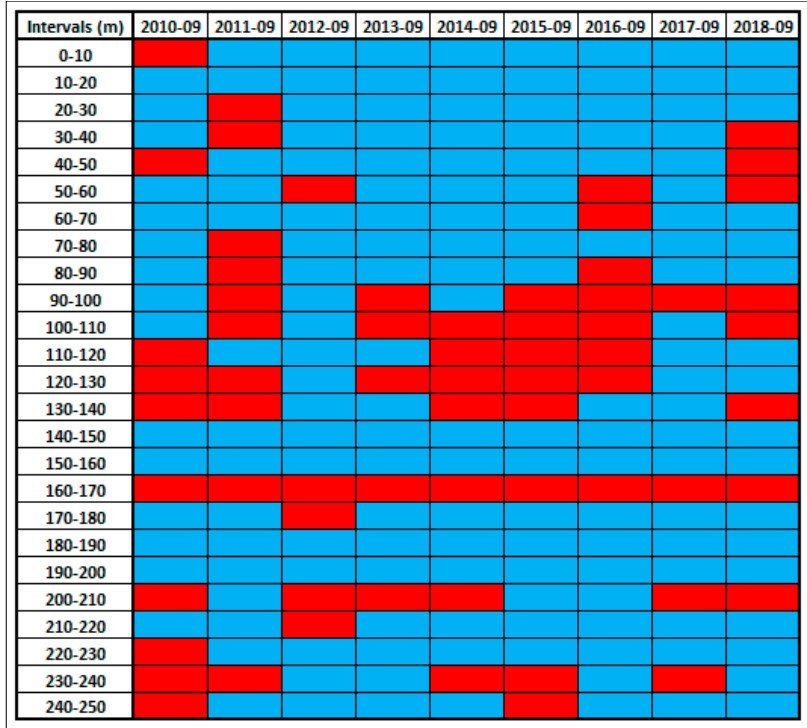

**Figure 8.** Annual evolution of profile 5 of cone B with respect to the year 2009. Red color indicates thinning and blue color indicates thickening.

### 4.3. Mathematical Prediction

#### 4.3.1. Result of Profile 3 of the Maximum Slope of Cone A

The sample $\{X_1, X_2, X_3, X_4, X_5, X_6, X_7, X_8\}$ is made up of the 8 curves belonging to the space of Hilbert $L^2_{[0,1]}$, and these are represented in Figure 9 with the prediction obtained for $\hat{X}_9$, which appears in red.

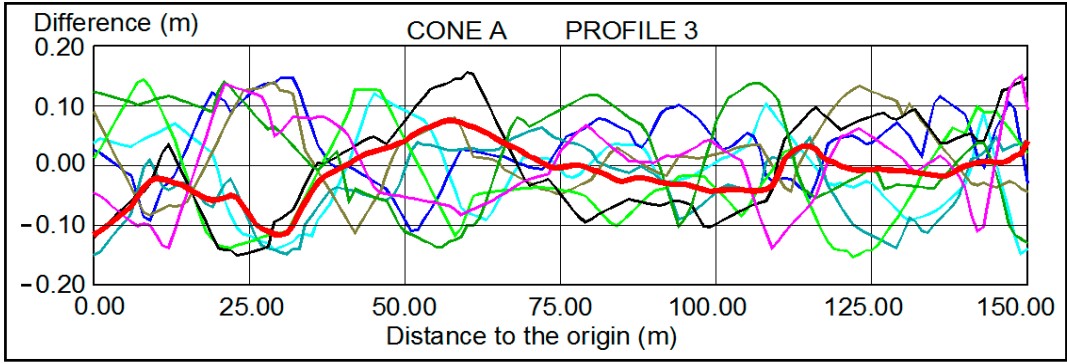

**Figure 9.** Predictive representation (in red) of profile 3 of Cone A. The rest of the curves belong to the space of Hilbert $L^2_{[0,1]}$.

If to $\hat{X}_9$ we add the mean $\frac{1}{8}\sum\limits_{j=1}^{8} X_j$, the prediction of the difference between 2018 and 2017 is obtained; therefore, by adding to it the data of 2017, we can make a prediction of the discrete values of 2018.

The field data are available for 2018. Figure 8 shows the difference between the graphs of the prediction for 2018 $(f(t))$ and the real data measured for the same period $(g(t))$ (Table 1). As can be seen in Figure 10, the difference for profile 3 of Cone A and the six profiles of Cone A in general were less than 15 cm.

**Table 1.** Numerical values for each distance to the origin of the profile of Figure 10, expressed in terms of the difference (m) between the predictive data and those obtained in the field for the year 2018.

| Distance | Difference | Distance | Difference | Distance | Difference | Distance | Difference | Distance | Difference | Distance | Difference |
|---|---|---|---|---|---|---|---|---|---|---|---|
| (m) | (m) | (m) | (m) | (m) | (m) | (m) | (m) | (m) | (m) | (m) | (m) |
| 1 | −0.138 | 26 | −0.042 | 51 | −0.059 | 76 | 0.019 | 101 | −0.065 | 126 | 0.058 |
| 2 | −0.133 | 27 | −0.026 | 52 | −0.050 | 77 | 0.026 | 102 | −0.057 | 127 | 0.028 |
| 3 | −0.128 | 28 | −0.029 | 53 | −0.049 | 78 | 0.037 | 103 | −0.054 | 128 | 0.011 |
| 4 | −0.125 | 29 | −0.030 | 54 | −0.047 | 79 | 0.048 | 104 | −0.040 | 129 | 0.000 |
| 5 | −0.118 | 30 | −0.048 | 55 | −0.034 | 80 | 0.047 | 105 | −0.027 | 130 | 0.004 |
| 6 | −0.113 | 31 | −0.046 | 56 | −0.019 | 81 | 0.041 | 106 | −0.014 | 131 | 0.011 |
| 7 | −0.107 | 32 | −0.036 | 57 | −0.004 | 82 | 0.030 | 107 | 0.000 | 132 | 0.005 |
| 8 | −0.080 | 33 | −0.018 | 58 | 0.008 | 83 | 0.022 | 108 | 0.002 | 133 | −0.010 |
| 9 | −0.065 | 34 | 0.001 | 59 | 0.012 | 84 | 0.012 | 109 | −0.008 | 134 | −0.050 |
| 10 | −0.062 | 35 | 0.010 | 60 | 0.010 | 85 | 0.000 | 110 | −0.005 | 135 | −0.091 |
| 11 | −0.060 | 36 | 0.016 | 61 | 0.009 | 86 | −0.012 | 111 | −0.012 | 136 | −0.113 |
| 12 | −0.057 | 37 | 0.021 | 62 | 0.006 | 87 | −0.017 | 112 | −0.028 | 137 | −0.125 |
| 13 | −0.066 | 38 | 0.035 | 63 | 0.005 | 88 | −0.028 | 113 | −0.039 | 138 | −0.135 |
| 14 | −0.076 | 39 | 0.047 | 64 | 0.006 | 89 | −0.055 | 114 | −0.037 | 139 | −0.136 |
| 15 | −0.088 | 40 | 0.017 | 65 | 0.013 | 90 | −0.090 | 115 | −0.017 | 140 | −0.138 |
| 16 | −0.106 | 41 | −0.030 | 66 | 0.023 | 91 | −0.123 | 116 | −0.013 | 141 | −0.143 |
| 17 | −0.124 | 42 | −0.013 | 67 | 0.032 | 92 | −0.130 | 117 | −0.004 | 142 | −0.148 |
| 18 | −0.141 | 43 | −0.020 | 68 | 0.030 | 93 | −0.134 | 118 | 0.005 | 143 | −0.147 |
| 19 | −0.156 | 44 | −0.030 | 69 | 0.005 | 94 | −0.138 | 119 | 0.018 | 144 | −0.131 |
| 20 | −0.129 | 45 | −0.043 | 70 | −0.018 | 95 | −0.133 | 120 | 0.030 | 145 | −0.111 |
| 21 | −0.121 | 46 | −0.061 | 71 | −0.007 | 96 | −0.128 | 121 | 0.047 | 146 | −0.095 |
| 22 | −0.098 | 47 | −0.077 | 72 | 0.004 | 97 | −0.121 | 122 | 0.055 | 147 | −0.089 |
| 23 | −0.084 | 48 | −0.080 | 73 | 0.010 | 98 | −0.112 | 123 | 0.061 | 148 | −0.091 |
| 24 | −0.069 | 49 | −0.079 | 74 | 0.012 | 99 | −0.107 | 124 | 0.065 | 149 | −0.081 |
| 25 | −0.055 | 50 | −0.073 | 75 | 0.017 | 100 | −0.081 | 125 | 0.069 | 150 | −0.066 |

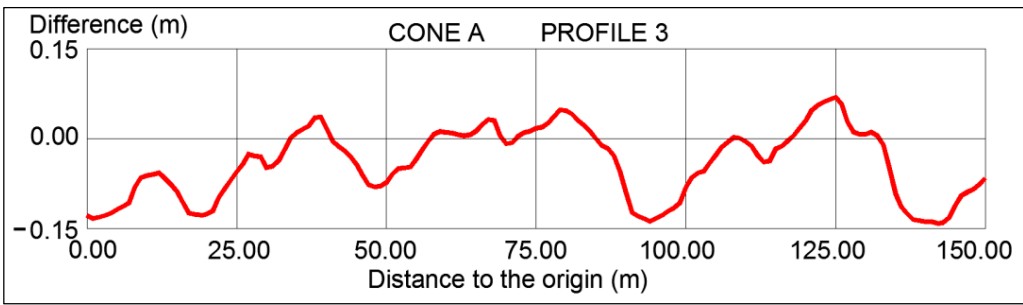

**Figure 10.** Representation of the difference between two results: The data are obtained as a prediction of 2018 and the real measured values of 2018 from profile 3 of Cone A.

There are different measurements of error between the curve obtained by prediction and reality. The commonest ones are the functional root-mean-squared error and functional mean absolute error:

- The functional root-mean-squared error obtained is

$$\frac{1}{151} \sqrt{\int_{1}^{151} (f(t) - g(t))^2 dt} = 0.081m \tag{3}$$

- The functional mean absolute error is

$$\frac{1}{151} \int_{1}^{151} (f(t) - g(t)) dt = 0.051m \tag{4}$$

### 4.3.2. Result of Profile 3 of the Maximum Slope of Cone B

Repeating the above process, the starting point is once more the sample $\{X_1, X_2, X_3, X_4, X_5, X_6, X_7, X_8\}$, which is made up of 8 curves represented in Figure 11, in which the prediction $\hat{X}_9$ obtained appears in red.

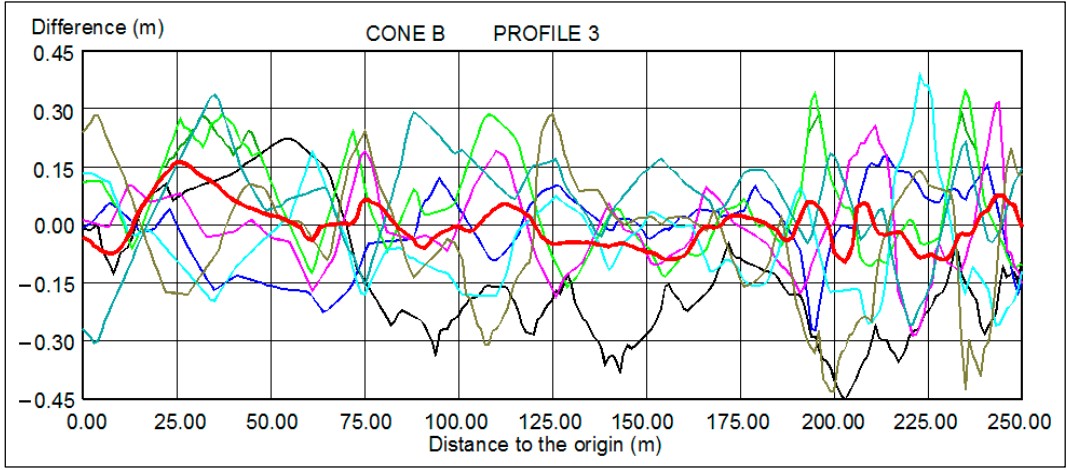

**Figure 11.** Predictive representation (in red) of profile 3 of Cone B. The rest of the curves belong to the space of Hilbert $L^2_{[0,1]}$.

Again, if we add the mean $\hat{X}_9$ to $\frac{1}{8} \sum_{j=1}^{8} X_j$, the prediction of the difference between the years 2018 and 2017 is obtained; therefore, by adding to it the data for 2017, we can make a prediction of the discrete values for 2018. Figure 12 shows the difference between the graphs of the prediction for

2018 ($f(t)$) and the real data measured for the same period ($g(t)$). As can be seen in Figure 12, the differences for profile 3 of Cone B were less than 30 cm.

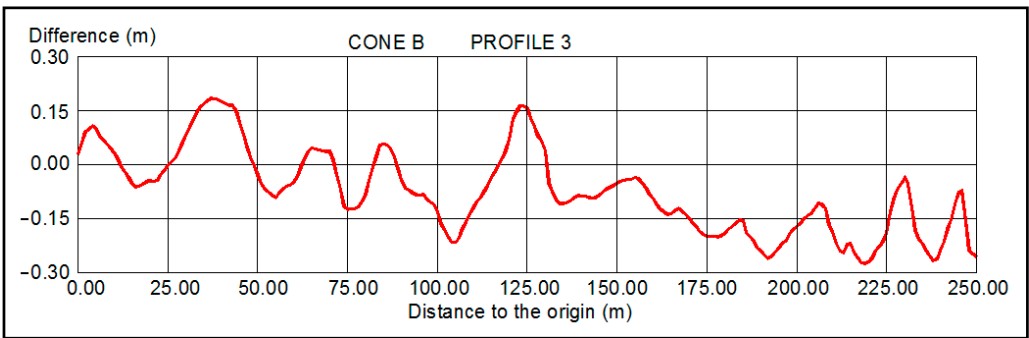

**Figure 12.** Representation of the difference between two results. The data are obtained as a prediction of 2018 and the real measured values of 2018 of profile 3, Cone B.

The same measurement errors as in the previous section are calculated once more between the prediction of the curve and the real one:

- The functional root-mean-squared error obtained is

$$\frac{1}{151} \sqrt{\int_1^{151} (f(t) - g(t))^2 dt} = 0.172m \tag{5}$$

- The functional mean absolute error is

$$\frac{1}{151} \int_1^{151} (f(t) - g(t)) dt = 0.110m \tag{6}$$

*4.4. Calculation of the Volume*

In the volumetric analysis for annual periods and the overall study period (2009–2018), both Cone A and Cone B showed discontinuous variability over the ten years analyzed and net differences in the redistribution of sediments (Figures 13 and 14). That is to say, the results show heterogeneity between two annual measurements. Therefore, it can be seen that there were no areas of continual thickening or thinning over time, but in areas in which there was thinning for some years, thickening took place in the following years. In general, annual volumetric variations were not greater than 0.3 m, except in areas of large rocks in Cone B and at the edges of the generation of the DEMs.

In order to understand the behavior of each debris cone, an independent analysis was required (Figures 13 and 14), the volumetric results of which are shown in Table 2.

4.4.1. Volume Calculation of Cone A

The surface of Cone A measured 14,812 m². Annual changes in volume showed an alternation between loss and gain of rock material. The annual analysis in the periods 2010–09 and 2013–12 revealed that a generalized loss of material took place. During the periods 2011–10 and 2014–13 there was a significant gain in material, which compensated for the preceding periods 2010–09 and 2013–12 (Table 2).

The remaining periods (2012–11, 2015–14, 2016–15, 2017–16, and 2018–17) showed an apparent equilibrium. From 2015 to 2018 there was a small gain of 394 m³.

The total of the period analyzed showed an insignificant increase in rock material of 515 m³. This points to a cone in a state of balance. The sediment of the cone's surface (14,812 m²) increased by 0.034 m/m² throughout its surface over the 10-year study period (2009–2018).

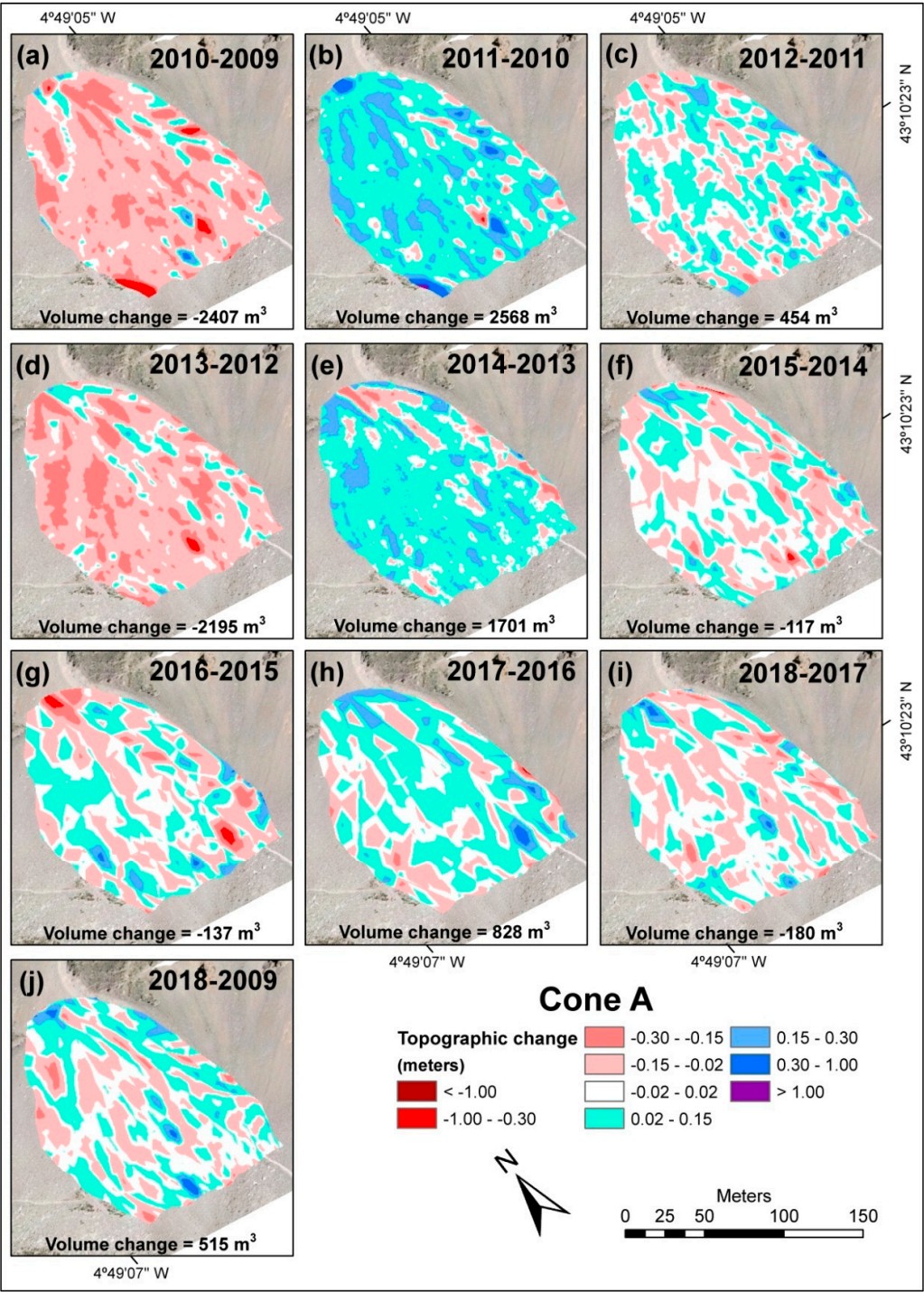

**Figure 13.** Annual (**a–i**) and total (**j**) differences (2018–2009) in volumes in Cone A.

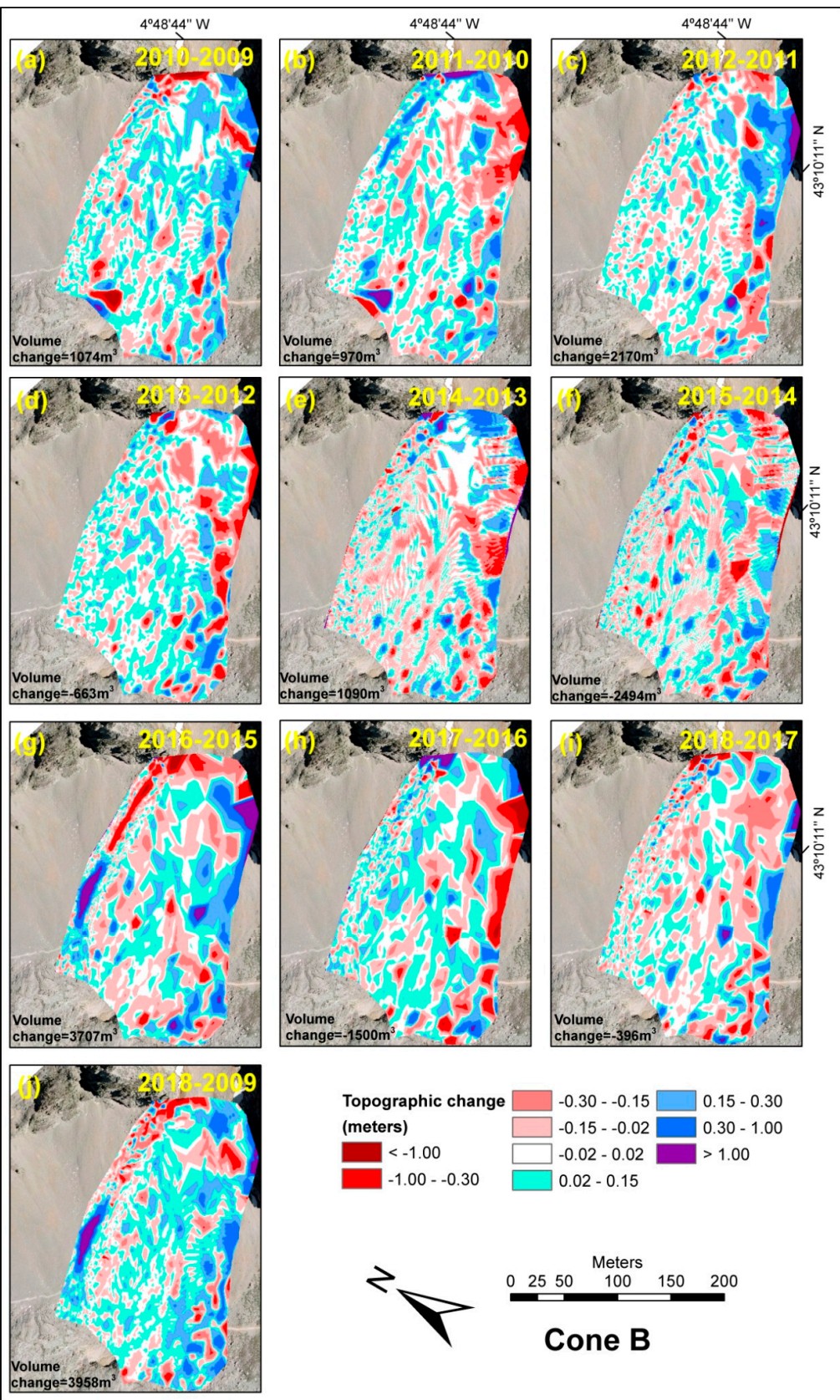

**Figure 14.** Annual differences (**a**–**i**) and total (**j**) difference (2018–2009) in volumes in Cone B.

**Table 2.** Annual and total changes (2018–2009) in volume in debris cones A and B.

| Years | Cone A (m$^3$) | Cone B (m$^3$) |
|---|---|---|
| 2010-2009 | −2407 | 1074 |
| 2011-2010 | 2568 | 970 |
| 2012-2011 | 454 | 2170 |
| 2013-2012 | −2195 | −663 |
| 2014-2013 | 1701 | 1090 |
| 2015-2014 | −117 | −2494 |
| 2016-2015 | −137 | 3707 |
| 2017-2016 | 828 | −1500 |
| 2018-2017 | −180 | −396 |
| 2018-2009 | 515 | 3958 |

### 4.4.2. Volume Calculation of Cone B

Cone B had a measured surface of 35,420 m$^2$. For each annual period there was no graphical predominance of thinning or thickening (Figure 14), but there was a proportional distribution of thickening and thinning throughout the cone. The analysis of the annual periods (Table 2) from 2009 to 2012 revealed a generalized thickening, the total value of which was 4214 m$^3$. From 2012 to 2017 thickening and thinning alternated and left an insignificant overall gain of 140 m$^3$ in the period. In 2009–2018, Cone B showed a tendency to thickening, 3958 m$^3$ in total. Therefore, if the accumulated 10-year gain is spread throughout the cone's surface, a gain of 0.11 m/m$^2$ is obtained, but it must be borne in mind that this cone contains the dynamic of the debris flow channel.

The debris flow channel behaved independently with respect to the rest of the cone: in the period 2011–10 it underwent significant thickening in its upper part, and in the period 2016–15 in its lower part. There was also thinning in the upper part of the flow channel in the period 2016–15. Therefore, in this period (2016–15) the transport of material from the upper to the lower areas of the flow channel was well established.

## 5. Discussion

Weathering and tectonic relaxation in the area of fracture and thrust are the factors that have determined the effectiveness of external agents on the walls and the feed of clasts to the slopes [37,38] since deglaciation around 15–12 ka years ago [49,52]. On the debris cones and taluses sediments are redistributed by rockfall, debris flow, creeping, and slides. The four most characteristic types of processes are subsidence, creeping, rolling, and rockfall [3,14,53–56], without ice working in the ground [27]. Moreover, debris flows constitute a remodeling element of the first order. In the cones analyzed debris lobes are dominant, although debris flows are the most active process in the removal of sediments, which is commonplace in alpine and polar debris taluses and cones [10,23,25,26]. These latter cases are associated with sudden melting, intense precipitation, or the combination of both, on already-saturated sediments by the persistence of snow patches until well into the summer.

The sizes of clasts are very heterogeneous. Although the mean size is 30 cm in the proximal and distal areas, there are large blocks (over a meter) in Cone B that affect the realization of the DEMs. The DEMs have a maximum difference of 15 cm for the generation of the same annual model with a grid measurement of 3 × 3 m. In the annual comparison of DEMs the maximum differences were 30 cm, although there were exceptions, as can be seen in Figures 13 and 14, mainly generated at the edges of the DEMs and the large blocks in Cone B.

Although each debris cone (A and B) was analyzed independently by means of maximum slope profiles, their behaviors were similar. We can, therefore, state the following:

- Debris cones maintain the equilibrium or maximum slope. Both measured cones have the same slope at the central portion (33.62° in Cone A, 33.54° in Cone B) (Figure 4). Evidently, the slope tends to be lesser in the distal portion;

- The wavy profile implies variations from one year to the next at the same point (Figure 5). The line of maximum slope sometimes shows alternative thinning and thickening areas (Figure 7). The periods of annual thickening and thinning show a minimum of five periods of each (accumulation and loss) for the total of nine periods studied (2010–2009, . . . , 2018–2017) (Figure 6). These facts show a wave behavior on the surface of the cones, particularly in Cone B;

- The analogical interpretation of the profiles of maximum slope analyzed (Figure 6) is as follows:

1. The heterogeneous variation in the profiles shows annual changes with thinning preceded or followed by years defined by thickening. The analysis of pairs of years (Figure 7) shows a compensating tendency, with periods alternating between thickening and thinning. These profiles are symmetrical around a value of 0;

2. There are not more than three consecutive periods (columns) of accumulation or loss for the same distance interval. This corroborates the observation made in the previous point, which is that there is no continuity over time of thickening or thinning in certain areas;

3. Figure 5 illustrates sequences of over five annual periods with continuous values of thickening "U" (purple lines) or thinning "D" (yellow lines). Sequence D (yellow line) in Figure 6 shows that the thinning at 80 and 90 m from the origin of the profile in the 2010–2009 period is found 10 m further back (now at 70–80 m) in the following period (2011–10). This process continues until 2015–2014 for a distance interval of 30–40 m. The wavy profile may be interpreted as creep on the surface, producing thickening and leaving thinning up and down. This process takes place continuously until the profile becomes stable (Figure 15).

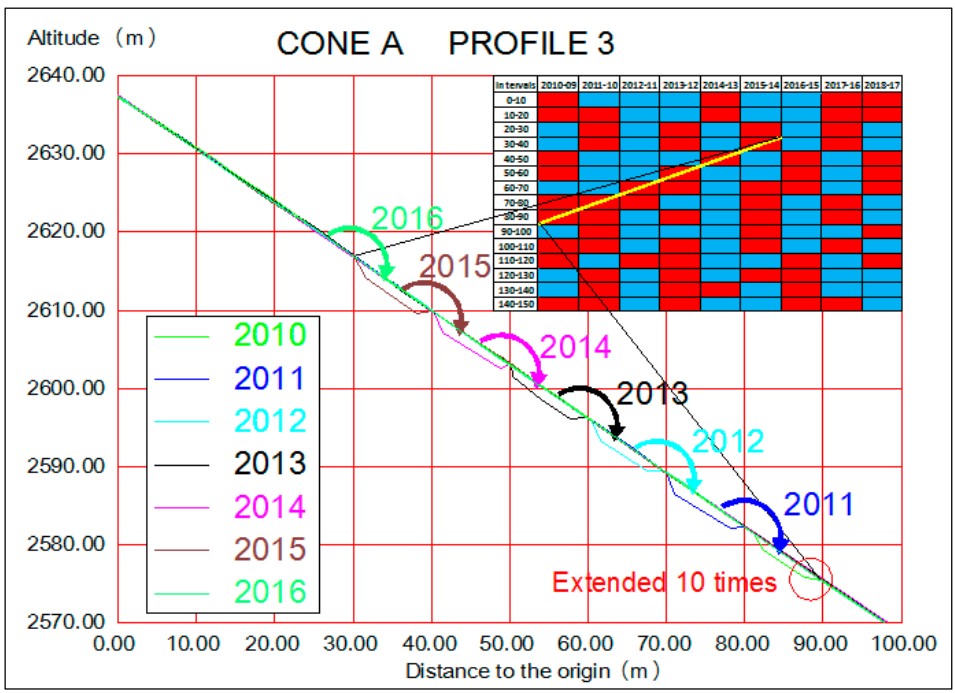

**Figure 15.** Profile indicating retreating thinning throughout a period of 6 years (2010–15). The effects of the retreat of the profile have been amplified ×10.

Upward thickening began where clasts were retained, which raised the level. The following year the retained clasts blocked sediments coming from above and altered and deformed the slope

with thickening taking place. This effect continued over the following years before stabilizing. Slope stabilization took place in 2015 (Figure 16).

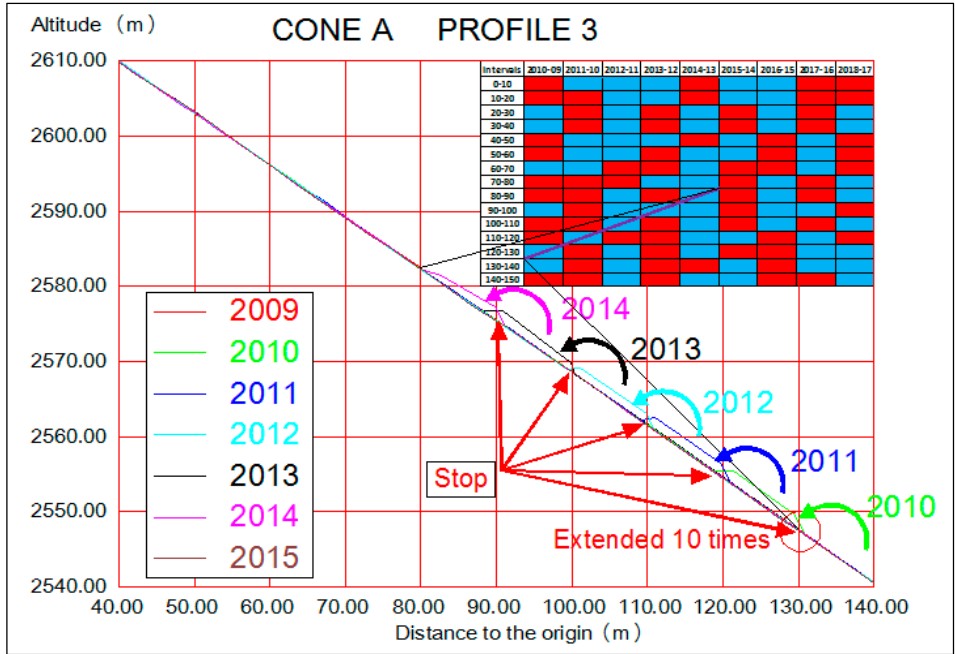

**Figure 16.** Profile indicating the thickening in retreat over a period of 5 years (2010–14). The thickening effect in the profile is amplified ×10.

The results of the linear analysis (maximum slope profiles) extrapolated to the whole of the surface of the debris cones permits the cones' volumetry to be known. The structure of Cone A remained stable during the period analyzed and its volume did not vary. Cone B has a debris flow channel included in the debris cone analysis as a whole. Debris flow has a low dynamic (0.1 events·a-1) when viewed against the whole of the massif (0.19 events a−1) [27]. Moreover, the cone contains large blocks, and when measuring the grids (3 × 3 m) of the DEMs, errors are generated, since this depends on the point where the measurement was made in each year. All of them indicate greater instability than Cone A, with a total of 3958 $m^3$ of clasts accumulated between 2009 and 2018. The accumulated material per $m^2$ is three times greater than in Cone A.

De Blasio and Sæter [56] and Bithell [32] suggest that rockfall may be the commonest process in the development of debris cones and talus, generating unstable slopes with surface clast movements. Creep generates a laminar structure in parallel layers. According to Serrano et al. [27], the presence of subsidence and transversal structures implies that there are two distinct areas: one with dominantly longitudinal processes and another with transversal deformations. The wavy profiles in both cones and throughout the profile points to the possible sliding on a basal laminar structure, as indicated by different authors using different methods [26,32,56,57], generating deformations, undulations, and transversal structures. These deformations reflect complex processes of transport and export of sediments in cascade sedimentary systems.

The prediction for 2018, by means of mathematical procedures using the annual data from the period 2009–2017 and the field measurement of 2018, reveals the difference between the predictive curve and that measured. This methodology was developed in profile 3 of Cone A and profile 3 of Cone B, leading to the following interpretations:

- The prediction of profile 3 of Cone B is less precise than that obtained in profile 3 of Cone A. In this case, the method is sensitive to some exceptional event (displacement of large blocks) in the distal portion, as can be seen in Figure 11. This lack of precision may also be due to the greater

difference between the curves (prediction and measured) in profile 3 of Cone B (Figure 11) than in profile 3 of Cone A (Figure 9). Figure 10 (Cone A) shows differences of +7 cm and −15 cm and Figure 12 (Cone B) differences of +16 cm and −27 cm;

- Cone A has a highly homometric texture with differences of less than 15 cm between the two curves (Figure 10). The overall average of the whole fit is 5 cm;
- The texture of Cone B is more heterometric, with large blocks mainly in the distal part, which has led to differences of 27 cm between the predictive curve for 2018 and the real measurement (Figure 12). The overall average of the whole fit is 11 cm;
- The curve of Cone A is of higher quality than that of Cone B, as the fit of Cone A is close to the precision of the equipment (2 cm). The curves of both cones are below the error produced in the generation of the DEM, which is 15 cm. It can be expected that with the increase in the number of field measurements, since only 10 years of measurements are available, the predictive curve will fit better to that measured. This must happen whenever there are no extraordinary events, such as large landslides or rockfalls, of which there were none during the studied period (2009–2018).

## 6. Conclusions

The Terrestrial Laser Scanning technique (TLS) has been shown to be highly efficient in monitoring annual surface and volumetric changes as well as the short- and medium-term trends in surface movement with suitable precision. This technique facilitates detailed knowledge of sediment transference processes in mountain slope systems. Independent data collection from two debris cones using TLS in the temperate high mountain (Picos de Europa, Spain) provided data of annual topographic changes and transfer of sediments from the walls to the cones in a cascade sediment system.

The TLS accuracy of this study was ±2 cm at each point measured. In the DEMs, generation differences of less than 10 cm were observed when the same cone was scanned twice on the same day. Only in some isolated cases were the differences around 15 cm or less (coinciding with clast size). In the distal area of Cone B there were blocks greater than 1 m, and differences between both DEMs, generated the same day, were greater than 15 cm and close to 1 m.

Maximum slope profiles were generated on the DEMs since the material supplied by the walls surrounding the debris cones followed the trajectory of the maximum slope line. Moreover, the surface analysis of the debris cones indicated heterogeneous behavior and was, therefore, difficult to evaluate using mathematical processes. Debris cones behave similarly, but each has its own particularities. Cone A presented a balance between sediment accumulation and transfer, a homometric texture, maximum slope profiles, and stable volumes, and its predictive model showed differences of less than ±5 cm. In Cone B, accumulation processes were dominant, the texture was heterometric with large blocks, and the comparison of the profiles revealed differences of ±45 cm (Figure 11), which in the areas with large blocks reached differences of over 1 m. This collectively means that the predictive model presents a worse fit in Cone B, with differences of ±11 cm.

The data collected over the ten-year period means the new surface structures in the debris cones can be observed together with the complex processes unrelated to ice and which involve a greater part of the cone than simply the surface layers. The undulated organization of the profiles points to a complexity of processes affirming that throughout the study area there are laminar displacements with differential velocities deforming the layers closest to the surface, with lesser deformations in the middle and upper areas.

The study of the debris cones is accompanied by functional data analysis techniques that lead to the prediction of their evolution. Although the annual data are collected form a discrete data set, they are values that belong to the curves or annual values that define the debris cones. In recent years statistical techniques have appeared that respect the continuous nature of phenomena of this kind: analysis and prediction of functional data.

Some of these techniques have been used to predict the values for 2018, fitting the model using the sample from previous years (2009–2017). Moreover, while the study was taking place, the real

values corresponding to 2018 were measured. Thanks to this, different error measurements between the predictive and the real values can be given with good results. It can, therefore, be established that this methodology is highly suitable for two debris cones in the Picos de Europa in Spain, and the results can be extrapolated to other debris cones of temperate high mountains (e.g., Southern Andes, the Alps, the Pyrenees, the Rocky Mountains, the Carpathian Mountains, the Atlas, the Pindus, the Caucasus, Pamir, or the Zagros). As the availability of historical values and data collection increase over the coming years, errors of prediction will diminish, as long as no high-intensity processes takes place, such as rockfalls, debris flows, or slope slides.

**Author Contributions:** Conceptualization, J.J.d.S.-B., E.S.; methodology, J.J.d.S.-B., M.L.-G., E.A.-P.; software, J.J.d.S.-B., M.L.-G., E.A.-P.; validation, J.J.d.S.-B., M.L.-G., E.A.-P.; formal analysis, J.J.d.S.-B., M.L.-G.; investigation, J.J.d.S.-B., E.S., M.L.-G., E.A.-P.; writing—draft, J.J.d.S.-B., E.S., M.L.-G., E.A.-P.; writing—review & editing, J.J.d.S.-B., E.S., M.L.-G., E.A.-P.; visualization, J.J.d.S.-B.; supervision, J.J.d.S.-B., E.S., M.L.-G., E.A.-P.; project administration, J.J.d.S.-B., E.S.; funding acquisition, J.J.d.S.-B., E.S. All authors have read and agreed to the published version of the manuscript.

**Funding:** This research was funded by European Regional Development Fund (ERDF) and the State Research Agency (AEI) of the Spanish Ministry of Economy and Competition, grant number TIN2016-76843-C4-2-R (AEI/FEDER, UE) and CGL2015-68144-R (MINECO/FEDER).

**Acknowledgments:** We wish to thank the Junta de Extremadura and the European Regional Development Fund (ERDF) for their support through the reference aid GR18053 for the research group NEXUS.

**Conflicts of Interest:** The authors declare no conflict of interest.

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
