# Peer review of "Modelling and Terrestrial Laser Scanning Methodology (2009–2018) on Debris Cones in Temperate High Mountains"

_remotesensing, doi:10.3390/rs12040632_

Round 1

Reviewer 1 Report

Brief summary:

It has been a pleasure to review the manuscript titled “Modelling and TLS application (2009‐2018) on debris cones in temperate high mountains “. The authors present an analysis of multitemporal Terrestrial Laser Scanner acquisitions in order to depict the evolution mechanisms of two debris cones.

Broad comments:

The analysis carried show some original points. Despite that, the authors should have broadened their work to all their dataset (i.e. the modelling is carried on only on the basis of two profiles). The prediction of “future displacements” seems to be weak: since it seems that for one case over two (cone B) the estimation is close to the error I would suggest to take into consideration all 12 profiles could give a wider sample to make some statistics on the results. In addition, the applicability and utility of these of results is not stresses in the text.

The volumetric estimation does not consider the error of measurements, I would suggest to do not consider volume variations caused by the height difference between +/- 0.15 m.

It would have been really interesting to use this dataset to locate the depth of the basal laminar structure (line 498 page 19) over which the sediment transport flows. It would have been possible using the minimal heights of all DEMs.

Lastly, it would be scientifically relevant to link volume variations or deposits income to meteorological forcing factors.

Follow some general comments divided according to the paragraphs of the manuscript:

“1 Introduction” and “2 Study area”

The English language of these sections need a revision, some sentences result quite hard to be followed.

“3. Data collection and methodology”

It is not described the way the authors carried repositioned the TLS station for each campaign and how the repositioning error would affect the measurements. I guess if the +/- 0.15 m error (page 6 line 226) of the DEM differences could be caused by TLS repositioning at each campaign

“3 Results”

“ 4.2 Calculation of the dynamic”
The “U” “D” labelling should not take into consideration variations that are below the measurement error (instrumental + repositioning error). The analysis should take into consideration data acquired at all 12 profiles.

“4.3 Mathematical prediction”
It should be clarified if the predicted profiles are within the measurement error.

“4.4. Calculation of the volume”
See comment of section “3 Data collection and methodology

Detailed comments are given in the attached pdf file.

Author Response

Dear review, I answer your questions:

It has been a pleasure to review the manuscript titled “Modelling and TLS application (20092018) on debris cones in temperate high mountains “. The authors present an analysis of multitemporal Terrestrial Laser Scanner acquisitions in order to depict the evolution mechanisms of two debris cones.

We are grateful to the referee for this comment for the recognition of the continued effort made over ten years on the same dates in the data collection process under difficult high mountain conditions carrying heavy equipment and in adverse weather conditions.

Broad comments:

The analysis carried show some original points. Despite that, the authors should have broadened their work to all their dataset (i.e. the modelling is carried on only on the basis of two profiles). The prediction of “future displacements” seems to be weak: since it seems that for one case over two (cone B) the estimation is close to the error I would suggest to take into consideration all 12 profiles could give a wider sample to make some statistics on the results. In addition, the applicability and utility of these of results is not stresses in the text.

The geometric analysis was performed in 12 profiles, 6 on each debris cone for all the years (2009-2018), as shown in Figure 3. The level for an interval of 1 metre in each profile for each year was established. From there the statistical analysis of the profiles was made. Reference is made in the article to the statistical calculations of profile 3 of each of the cones.

As indicated in the article (lines 211-213) the predictive analysis was made on two profiles (one for each debris cone): “This methodology may be applied to any of the profiles for which data have been taken. For example, it was applied to profile 3 of cone A (Figures 5 and 6) and profile 3 of cone B, and on these the mathematical modelling was developed”.

These two profiles were selected because in one of them the data were quite close (with less variability) and in the other there was more dispersion. With the remaining profiles predictions were obtained with intermediate errors between the two cases studied.

All the data and results cannot be supplied in the article, and so reference is continually made to the calculations of different profiles in each debris cone. The results, discussion and conclusions do not refer to profile 3 of each debris cone alone, rather to the calculations of the 12 profiles. The geometric analysis was performed for the 12 profiles, that is, six profiles for each debris cone, as shown in Figure 3, and for all years (2009-2018).

The volumetric estimation does not consider the error of measurements, I would suggest to do not consider volume variations caused by the height difference between +/- 0.15 m.

Among the intervals assigned In figures 13 and 14 are: +/- 0.02 cm and +/- 15 cm. The reason for these intervals is that the error of the instruments is 2 cm, and therefore it is within this interval that there is no displacement. Also, the interval of 15 cm was considered as it was understood that in a single survey it is the maximum value that we have of uncertainty in the generation of the DEMs. These data are indicated in the article, specifically in lines 58-159: “The mean differences between the two models were 10 cm, and differences did not surpass values of 15 cm in any case except in the distal part of cone B”. Therefore, the maximum value is 15 cm, but the differences between the two models are of less than 15 cm.

As the referee states, this modification has been made eliminating the 2 cm interval from the calculation of the volumes in both of the figures 13 and 14 and in the text (lines 224-227). The intervals indicated by the referee have been changed, but we believe that the interpretation was better as it was previously. The interval +/- 15 cm now appears in white, and it gives the impression that there is no dynamism in the debris cone.

We must remember that 15 cm is the maximum value in the comparison of two DEMs in the same year, but that the values in general are below 10 cm.

We are sending the referee the volumetric and graphic calculation (figures 13 and 14) in both form so that he/she may select the one offering best option.

It would have been really interesting to use this dataset to locate the depth of the basal laminar structure (line 498 page 19) over which the sediment transport flows. It would have been possible using the minimal heights of all DEMs.

The only way of knowing the internal structure and depth of basal laminar structure is by GPR, but the slope and blocky texture made it impossible to get good results. The best way is to work on the snow, but the environmental conditions and snow avalanche risks did not permit such work during the periods with snow cover. We consider that minimal heights of all DEMs fail to indicate the possible basal laminar flow because the deposit is so much deeper than the wave features.

Lastly, it would be scientifically relevant to link volume variations or deposits income to meteorological forcing factors.

We agree with the referee's assessment, and it would be interesting to compare the meteorological forcing factor and volume variations. Nevertheless the available weather stations are far from the study area and the emplacement very different from wall and channel environments. We therefore did not conduct a study on climatological conditions during the years of the study.

Follow some general comments divided according to the paragraphs of the manuscript:

“1 Introduction” and “2 Study area”n

The English language of these sections need a revision, some sentences result quite hard to be followed.

A native translator translated the article, although a full review of the article has been made again.

If this article is accepted for publication, a grammatical revision will be made by MDPI's English editing service.

“3. Data collection and methodology”

It is not described the way the authors carried repositioned the TLS station for each campaign and how the repositioning error would affect the measurements. I guess if the +/- 0.15 m error (page 6 line 226) of the DEM differences could be caused by TLS repositioning at each campaign.

This is true. In the generation of the DEMs, the value of 0.15 m is the maximum difference generated in the same cone and in the same observation survey. Therefore, the error of the TLS positioning would be included within this value. The error of TLS positioning is 2 cm (TLS stationing error and error of the measurement of instrument distance).

As indicated in the text the difference of 0.15 m is due to the fact that in the measurement of each DEM, the points measured are not the same and generate these maximum differences, this value being similar to the size of the clasts.

“3 Results”

“4.2 Calculation of the dynamic”. The “U” “D” labelling should not take into consideration variations that are below the measurement error (instrumental + repositioning error). The analysis should take into consideration data acquired at all 12 profiles.

The referee’s comment is correct. Below the maximum differences of 15 cm it cannot be confirmed whether the profile increases (U) or decreases (D) for a certain interval. But if these intervals were not catalogued with a value of U or D many of the spaces would remain undefined and there would not be any continuity in the qualitative values (Figure 5 and 6). The same thing happens with intervals of 10 metres (Figure 5), in which a situation of a value of U changes to a value of D and vice versa.

This part is very complicated to define given that the variations between two years are very small (this is common to all the profiles and in both cones), but the same thing happens in the overall period of 10 years. For this reason, we have indicated in the article that there is an equilibrium slope in each cone, and also a tendency towards an ondulating dynamic, as shown by the statistical analysis. This analysis confirms that there is a movement with a certain tendency, and for this reason the position can be predicted quite reliably throughout the profile for a later year, as long as a long data sequence is available (10 years).

“4.3 Mathematical prediction”. It should be clarified if the predicted profiles are within the measurement error.

As indicated in lines 320-322: “As can be seen in Figure 10, the difference for profile 3 of Cone A and the six profiles of Cone A in general are less than 15 cm”. These statistical values are below the error in the generation of DEMs in a same survey.

With regard to the profile of cone B, in lines 344-345 we state: “As can be seen in Figure 12, the difference for profile 3 of cone B is less than 30 cm”, but in general for the profile the predictive values are less than 15 cm, except in the lower part, where there are larger blocks of clasts. This is described in lines 508-514: “The prediction of profile 3 of cone B is less precise (the errors are larger) than that obtained in profile 3 of cone A. In this case, the method is sensitive to some exceptional event (large stone blocks), mainly at the end of the profiles, as can be seen in Figure 11. This lack of precision may also be due to the greater difference between the curves (prediction and measured) in profile 3 of cone B (Figure 11) than in profile 3 of cone A (Figure 9). This can be seen in Figure 10 (corresponding to cone A) with differences of + 7 cm and -15 cm and Figure 12 (cone B) with differences of + 16 cm and - 27 cm”.

We wish to point out that a predictive method obtains estimates or predictions of future values based on historical data available. Therefore, the greater the historical series, the better the prediction if the method has been suitably chosen for the type of sample in question.

“4.4. Calculation of the volume”. See comment of section 3 Data collection and methodology

Comments are in the attached pdf.

New figures 13 and 14. The difference of +/- 15 cm is not considered for the calculation:

Figure 13 and 14:

Excuse me, figures 13 and 14 are impossible to attach. Please ask the editor.

Reviewer 2 Report

I suggest change the title as methodology in accord with the conclusions. Could be reduce the discussion, the conclusion are relevant. 

Author Response

Dear reviewer, I answer your questions:

I suggest change the title as methodology in accord with the conclusions. Could be reduce the discussion, the conclusion are relevant.

As the reviewer indicates, in the title “application” has been changed by “methodology” and the discussion part has been reduced.

The translation has been done by an English person and, in response to the comments of the reviewers, the entire document has been revised again. If the article is accepted, a grammar review will be done by MDPI's English editing service.

Reviewer 3 Report

To the authors:

This work is very well done, and I would recommend medium revisions. There are quite a few grammatical errors, some of which I pointed out. If the authors aren’t native English speakers, I might recommend having someone look the manuscript over. The writing is good, there’s just quite a few long sentences. My greatest concern with the manuscript is the small sample size (i.e., only two cones). You state that you can “extrapolate the modeling” to debris cones in other temperate high mountain regions. This seems questionable to me, especially since both of the cones are in the same area. I understand the limitations of sample size (or perhaps I’m just misunderstanding), so even an explanation of why this sample size is reasonable, should suffice. Beyond that, I’d consider redoing some of the figures, so that people that are color blind aren’t affected by the green/red mix. It’s not always easy to do that, but where you can (e.g., the middle and bottom figure of Figure 7) I would recommend that.

All that said, this is an exciting paper and I commend all of the authors for their work here. The following is a few notes by page and line number.

Introduction, page 2, line 46-48: This sentence is confusing. I’m not sure what you mean by it. What does the but refer to? Creep, rolling, … have been studied, but what hasn’t?

Introduction, page1, paragraph 2. This is good material, but sentences are really long. Could you break them up? As a reader, the long sentances are hard to follow.

Figure 1. The map here is a quite dark. Could you lighten it and perhaps show more detail about the area? Perhaps a zoomed in map of the area with an inset map with the general location in Spain.

Introduction, page3, top paragraph, line 85. It concerns me that because two formulations are the same that this could be used to extrapolate it to debris cones in other temperate high mountain regions. This seems a rather small sample size. Particularly, since Cone A and Cone B are in the same region.

The study area description is really good.

A note on figures in the results. You use a lot of green and red, which is often problematic for people who are color blind. This is often difficult to recifty, but when possible, I’d suggest a different color scheme when red and green are your choices. Blue and red are better, as that type of color blindness is less common.

Results, page 10, line 319. You write, “ Figure 8 show …” it should read, “Figure 8 shows ….”

Discussion, page 18, lines 465-468. The wording of this sentence is confusing.

Discussion, page 18, lines 468 and 469. Both sentences begin with “this.” In both cases, it is unclear what “this” is referring to. In grammar, “this” at the beginning of sentences is often an unclear antecedent.

Discussion, page 19, lines 474 to 475. The wording in this sentence is confusing.

Conclusion, page 20, line 536 to 539 (second sentence of paragraph). This sentence is long and confusing.

Author Response

Dear reviewer, I answer your questions:

This work is very well done, and I would recommend medium revisions. There are quite a few grammatical errors, some of which I pointed out. If the authors aren’t native English speakers, I might recommend having someone look the manuscript over. The writing is good, there’s just quite a few long sentences. My greatest concern with the manuscript is the small sample size (i.e., only two cones). You state that you can “extrapolate the modeling” to debris cones in other temperate high mountain regions. This seems questionable to me, especially since both of the cones are in the same area. I understand the limitations of sample size (or perhaps I’m just misunderstanding), so even an explanation of why this sample size is reasonable, should suffice. Beyond that, I’d consider redoing some of the figures, so that people that are color blind aren’t affected by the green/red mix. It’s not always easy to do that, but where you can (e.g., the middle and bottom figure of Figure 7) I would recommend that.

A native translator translated the article, although a full review of the article has been made again. If the article is accepted for publication, a grammatical review will be made by MDPI's English editing service.

Lines 86 and 87: The results of this article may have been overrated with this phrase, such as the possibility of extrapolating them to other parts of the world. The phrase has therefore been edited out. We have analyzed twelve profiles in the two cones (six in each debris cone) located in the same geographical area and climatic conditions.

High mountain conditions (displacement at the foot with heavy equipment, changing climatological conditions, time limitations for making measurements during the working day) reduce the chances of getting the desired number of samples for the study. The study was carried out on these two debris cones on the same dates over 10 consecutive years (2009-2018). Samples from the 12 profiles were observed and two of them were chosen for prediction by statistical techniques: one in which the data were quite similar (with less variability) and another in which they were more disperse. Predictions were obtained using the remaining profiles with errors somewhere between the two cases studied.

Figure 7: The green colour has been substituted for black (2015-14).

All that said, this is an exciting paper and I commend all of the authors for their work here. The following is a few notes by page and line number.

Introduction, page 2, line 46-48: This sentence is confusing. I’m not sure what you mean by it. What does the but refer to? Creep, rolling, … have been studied, but what hasn’t?

The sentence has been changed. It now states: “The main, processes analyzed that are involved in the debris dynamic are rockfall, snow avalanches and debris flows, but surface processes such as creep, rolling, solifluction, physical and chemical weathering, and surface runoff are also present [5,6,11–16]”.

Introduction, page1, paragraph 2. This is good material, but sentences are really long. Could you break them up? As a reader, the long sentances are hard to follow.

The sentence has been changed. It now states: The data obtained using different methods for volume and structure measurements of debris cones and taluses may be biased due to the complexity of the system feeding the whole area. Walls and channels in turn a source of materials for transport towards distal areas, with processes that remodel the surface and alter its structure in a cascade sedimentary system [18–22]. These processes are not only geomorphological, such as nivation, debris flows, solifluction or gelifluction, but also include plant colonization, trampling by animals, and anthropogenic intervention, such as paths and infrastructures, which all contribute to the alteration with high spatial and temporal variability [23,24]. For the study of debris flow, debris talus and cone volumes in the temperate mountains of the world, such as the Southern Andes, the Alps, the Pyrenees, the Rocky Mountains, the Carpathian Mountains, the Atlas, the Pindus, the Caucasus, Pamir or the Zagros, the great potential of remote methods such as Terrestrial Laser Scanning (TLS) and photogrammetry with Unmanned Aerial Vehicles (UAV) is well known [25], and the new observation and recording techniques provide more detailed knowledge of the dynamic of taluses and cones [26,27].

Figure 1. The map here is a quite dark. Could you lighten it and perhaps show more detail about the area? Perhaps a zoomed in map of the area with an inset map with the general location in Spain.

Figure 1 shows the geographical map. Nevertheless, a new map has been included with details of the altitudes in the area, as the referee recommends.

Introduction, page3, top paragraph, line 85. It concerns me that because two formulations are the same that this could be used to extrapolate it to debris cones in other temperate high mountain regions. This seems a rather small sample size. Particularly, since Cone A and Cone B are in the same region.

The referee is correct. Lines 86-87 have now been omitted: “The formulation obtained for the two debris cones is identical, and so we can extrapolate it to debris cones in other temperate high mountain regions”.

The study area description is really good.

Thanks.

A note on figures in the results. You use a lot of green and red, which is often problematic for people who are color blind. This is often difficult to recifty, but when possible, I’d suggest a different color scheme when red and green are your choices. Blue and red are better, as that type of color blindness is less common.

The colours have been changed, as the referee requests. In Figure 7 the green colour has been substituted for black.

Results, page 10, line 319. You write, “ Figure 8 show …” it should read, “Figure 8 shows ….”

This has now been changed.

Discussion, page 18, lines 465-468. The wording of this sentence is confusing. Discussion, page 18, lines 468 and 469. Both sentences begin with “this.” In both cases, it is unclear what “this” is referring to. In grammar, “this” at the beginning of sentences is often an unclear antecedent.

The sentence has now been rephrased: Upward thickening begins where clasts are retained, which raises the level. The following year the retained clasts block sediments coming from above and alter and deform the slope with thickening taking place. This effect continues over the following years before stabilizing. Slope stabilization took place in 2015 (Figure 16).

Discussion, page 19, lines 474 to 475. The wording in this sentence is confusing.

The sentence has been eliminated.

Conclusion, page 20, line 536 to 539 (second sentence of paragraph). This sentence is long and confusing.

The sentence has been rephrased: The TLS accuracy of this study was ±2 cm at each point measured. In the DEMs generation differences of less than 10 cm were observed when the same cone was scanned twice on the same day. Only in some isolated cases were the differences around 15 cm or less (coinciding with clast size). In the distal area of cone B there are blocks greater than 1 m and differences between both DEMs, generated the same day are greater than 15 cm and close to 1 m.

Reviewer 4 Report

The topic of this manuscript is interesting and valuable. To my opinion, however, there are many problems need to be solved. As a whole, a major revision is suggested to this paper. Please rewritten this paper, otherwise, a rejection is suggested. Several main reasons are provided as below:

The abbreviation should not be written in the title. We do not know what is TLS The novel findings of this study are not described well in the part of “Abstract”. The written English of this paper should be improved significantly. The writing logic and readability of the paper also need to be improved. The description of Figure 4 is not clear. The part of “Mathematical modeling” and some other parts are very messy. One paragraph should not be too short. The mathematical notation should be edited by the “MathType”. Some figures should be merged or deleted. It is not necessary to provide the abbreviation in the conclusion part, suggesting that the essay writing abilities of the authors are poor. The conclusions of this study are described too much. You should refine important conclusions, not discussions and descriptions.

Author Response

Dear reviewer, I answer your questions:

The topic of this manuscript is interesting and valuable. To my opinion, however, there are many problems need to be solved. As a whole, a major revision is suggested to this paper. Please rewritten this paper, otherwise, a rejection is suggested. Several main reasons are provided as below:

The abbreviation should not be written in the title. We do not know what is TLS.

The meaning of TLS is Terrestrial Laser Scanning. As the reviewer indicates, the title has been changed.

The novel findings of this study are not described well in the part of “Abstract”.

The abstract of the article indicates: “In addition, the statistical predictive value for position (Z) in the year 2018 was calculated for the same planimetric position (X,Y) throughout the profiles of maximum slopes. To do so, the real field data from 2009-2017 were interpolated and used to form a sample of curves. These curves are interpreted as the realization of a functional random variable that can be predicted using statistical techniques. The predictive curve obtained was compared with the 2018 field data. The results of both coordinates (Z), the real field data and the statistical data are coherent within the margin of error of the data collection”.

We think that the novelty of the article is that we demonstrate that debris cones are not static, therefore there is dynamics. The movement can be predicted by statistical techniques. But, to apply this methodology it is necessary to have a large series of data and it continues over time. We have taken data for 10 years (2009-2018) and are currently continuing to collect data to continue demonstrating what is indicated in this article.

The written English of this paper should be improved significantly. The writing logic and readability of the paper also need to be improved.

A native translator translated the article, although a full review of the article has been made again.

If this article is accepted for publication, a grammatical revision will be made by MDPI's English editing service.

The description of Figure 4 is not clear.

The article indicates: “Figure 4. Representation of the annual evolution (2009-2018) of profile 2 of cone A. The blue box is amplified x100 times with respect to the original profile (black box)”.

It has been changed to the following sentence: “Figure 4. Representation of the annual evolution (2009-2018) of profile 2 of cone A (see Figure 3). The blue box is amplified x100 times with respect to the original profile (red box)”.

The part of “Mathematical modeling” and some other parts are very messy. One paragraph should not be too short. The mathematical notation should be edited by the “MathType”.

As the reviewer indicates, the formulas have been edited.

Some figures should be merged or deleted. It is not necessary to provide the abbreviation in the conclusion part, suggesting that the essay writing abilities of the authors are poor.

Some figures have been improved with the indications of other reviewers. Grammar will be improved when the grammar review is done by MDPI's English editing service.

The conclusions of this study are described too much. You should refine important conclusions, not discussions and descriptions.

The discussion section is very extensive and a new writing has been made

Round 2

Reviewer 1 Report

Dear Authors,

many thanks for having taken in consideration all my notes by presenting detailed responses to all of them.

Unluckily, I didn’t have the chance to view the second version of figures 13 and 14 (without the ±0.02m interval) but I think I can imagine them. Please keep the ±0.02 m interval in figures 13 and 14 and following tables.

I still think that topographic variations below 0.15 meters must be carefully taken into consideration when you derive your conclusions and that this range of measures, that actually reflects majority of topographic changes useful to your dynamic model, must be handed with caution.

That said I thank again for your responses to my comments and I will recommend the editorial office to publish your work.

Author Response

Dear reviewer:
We appreciate your comments.
We keep figures 13 and 14 as were initially and I send you the modifications (Figures 13 and 14) you indicated in your previous review.
Greetings

Reviewer 4 Report

This paper can be published after English Editing.

Author Response

Dear reviewer:

Grammatical revision will be made by MDPI's English editing service.

Best regards.